# Synergistic apoptotic effects in cancer cells by the combination of CLK and Bcl-2 family inhibitors

**Aiko Murai\*, Shunsuke Ebara, Satoshi Sasaki, Tomohiro Ohashi, Tohru Miyazaki, Toshiyuki Nomura, Shinsuke Araki⊙\***

Research, Takeda Pharmaceutical Company Limited, Fujisawa, Kanagawa, Japan

\* aiko.murai@takeda.com (AM); shinsuke.araki@takeda.com (SA)

**Data Availability Statement:** All relevant data are within the manuscript and its Supporting Information files.

## Abstract

Emerging evidence indicates that alternative splicing plays a critical role in cancer progression through abnormal expression or mutation of splicing factors. Small-molecule splicing modulators have recently attracted considerable attention as a novel class of cancer therapeutics. CDC-like kinases (CLKs) are central to exon recognition in mRNA splicing and CLK inhibitors exhibit anti-tumour activities. Most importantly, molecular mechanism-based combination strategies for cancer therapy must be considered. However, it remains unclear whether CLK inhibitors modulate expression and splicing of apoptosis-related genes, and whether CLK inhibitors enhance cytotoxicity in combination with apoptosis inducers. Here we report an appropriate mechanism-based drug combination approach. Unexpectedly, we found that the CLK inhibitor T3 rapidly induced apoptosis in A2780 cells and G2/M cell cycle arrest in HCT116 cells. Regardless of the different phenotypes of the two cancer cell types, T3 decreased the levels of anti-apoptotic proteins (cIAP1, cIAP2, XIAP, cFLIP and Mcl-1) for a short period of exposure and altered the splicing of the anti-apoptotic *MCL1L* and *CFLAR* isoform in A2780 and HCT116 cells. In contrast, other members of the Bcl-2 family (i.e., Bcl-xL and Bcl-2) were resistant to T3-induced expression and splicing modulation. T3 and a Bcl-xL/Bcl-2 inhibitor synergistically induced apoptosis. Taken together, the use of a CLK inhibitor is a novel therapeutic approach to sensitise cancer cells to Bcl-xL/Bcl-2 inhibitors.

## Introduction

Alternative pre-mRNA splicing is a fundamental mechanism that generates multiple mRNAs from a single gene via a mechanism that is tightly regulated to generate proteomic diversity sufficient to maintain physiological homeostasis and processes [1–4]. Dysregulation of alternative splicing leads to the generation of aberrant protein isoforms that contribute to various diseases, including neurodegenerative diseases, muscular dystrophies and various cancers [5]. Especially, specific aberrant splicing of various transcripts, such as Bcl-xL, Cyclin D1, CD44, and VEGF, is known to promote tumour survival and growth, as well as resistance to apoptosis as a consequence of the abnormal expression or mutation of splicing factors [6–10]. Recent

**Funding:** Takeda Pharmaceutical Company Limited provided support in the form of salaries for authors but did not have any additional role in the study design, data collection and analysis, decision to publish, or preparation of the manuscript. The specific roles of these authors are articulated in the 'author contributions' section.

**Competing interests:** In accordance with the journal's policy regarding conflicts of interest: AM, SE, SS, TO, TM, TN and SA are/were employees of Takeda Pharmaceutical Co. Ltd. This does not alter the authors' adherence to all the PLOS ONE policies on sharing data and materials.

whole genome and RNA sequence analyses across multiple haematologic and solid tumour types have identified mutually exclusive somatic mutations that affect key components of the splicing machinery, such as SF3B1, U2AF1, SRSF2 and U2AF35 [8–10].

Compounds targeting the spliceosome machinery have been identified as potential targets in cancer therapy. H3B-8800 is an orally administered modulator of the SF3b complex that has potent anti-tumour activity against spliceosome-mutant tumour cells [11]. In addition, the oncogenic roles of CDC-like kinases (CLKs) have been identified in cancers of the breast and kidney [12,13]. CLK inhibitors also have anti-tumour activities that occur through the modulation of factors involved in cancer-associated splicing that are aberrantly expressed by cancer cells [14,15]. T3 is highly selective to CLKs and the potent small molecule compounds using kinase panel and comprehensive RNA-seq analysis between silencing of CLKs and T3 treatment [15].

The results of some clinical trials have shown that various splicing modulators have potential value as a novel class of anti-tumour agents. It is crucial to consider the molecular mechanisms underlying therapeutic strategies for cancer treatment. Especially, the use of a combination of cancer drugs is particularly beneficial due to the utilisation of a mechanism-based approach, since the efficacy of a single anticancer agent is currently limited.

The SF3B1 inhibitor E7107 in combination with Bcl-xL/Bcl-2 inhibitors enhances cytotoxicity to cancer cells based on the evidence that E7107 alters splicing of *MCL1*, but not that of *BCL2L1* [16]. Furthermore, silencing of the splicing factor SF3B1 or SRSF1 has been shown to induce splicing alteration of *MCL1*, which subsequently reduces Mcl-1L protein expression by chronic lymphocytic leukaemia cells [17–19]. Although recent evidence suggests that Myc-amplified cancer cells are sensitive to the CLK inhibitor T-025 [20], little is known about whether CLK inhibitors are involved in the expression and splicing modulation of apoptosis-related genes to inhibit cell growth and induce apoptosis.

The results of the present study clarified that the CLK inhibitor T3 led to decreased levels of anti-apoptotic proteins, IAP family (cIAP1, cIAP2, XIAP), cFLIP, and Mcl-1for a short period of exposure, while Bcl-xL and Bcl-2 proteins were resistant. T3 altered the splicing of anti-apoptotic *MCL1L* and *CFLAR* isoform, while *BCL2L1* and *BCL2* were resistant to T3-induced splicing modulation. Thus, the combination of Bcl-xL/Bcl-2 inhibitors and T3 enhanced apoptosis of cancer cells. These data suggest that the splicing modulator T3 in combination with Bcl-xL/Bcl-2 inhibitors may be valuable to induce synergistic apoptosis as a novel cancer therapeutic strategy.

## Materials and methods

### Cell culture

Human colorectal cancer HCT116 cells and human ovarian cancer A2780 cells were obtained from the American Type Culture Collection (Manassas, VA, USA). HCT116 cells were maintained in McCoy's 5a growth medium (Thermo Fisher Scientific, Waltham, MA, USA) and A2780 cells were maintained in RPMI-1640 growth medium (Thermo Fisher Scientific). Both media were supplemented with 10% fetal bovine serum (Thermo Fisher Scientific), penicillin (10,000 U/mL; Thermo Fisher Scientific) and streptomycin (10,000 U/mL; Thermo Fisher Scientific). All cells were maintained in a humidified 37°C incubator with 5% $CO_2$ and were routinely tested for the absence of mycoplasma infection using the MycoAlert Mycoplasma Detection Kit (Lonza, Basel, Switzerland).

### Compounds

[4-(2-methyl-1-(4-methylpiperazin-1-yl)-1-oxopropan-2-yl)-*N*-(6-(pyridin-4-yl)imidazo [1,2-*a*]pyridin-2-yl)benzamide] (T3) was synthesised and purified as described previously [15]. [N-

(8-fluoro-6-(pyridin-4-yl)imidazo[1,2-a]pyridin-2-yl)-4-(2-methyl-1-(4- methylpiperazin-1-yl)-1-oxopropan-2-yl)benzamide trihydrochloride] (T3-1) was synthesised and purified as shown in Supporting Information (S1 File). The Bcl-2 family protein inhibitor ABT-263 was purchased from MedChemExpress (HY-10087; Monmouth Junction, NJ, USA). The pan-caspase inhibitor Z-VAD-FMK was purchased from R&D Systems (FMK001; Minneapolis, MN, USA) and used at 25 μM in all the experiments described in the present study.

## Cell growth assay

A2780 and HCT116 cells (each $3.0 \times 10^3$ per well) were seeded into the wells of 96-well plates and incubated overnight under the conditions described above. The following day, the diluted compounds or dimethyl sulfoxide (DMSO) as a control were added to the culture media. After 72 h of incubation, the cells were assessed using the CellTiter-GloLuminescent Cell Viability Assay (Promega Corporation, Madison, WI, USA), in accordance with the manufacturer's instructions. The chemiluminescence of each well was measured using an ARVO X3 Micro-plate Reader (PerkinElmer, Inc., Waltham, MA, USA). $GI_{50}$ values (the drug concentration that inhibits cell growth by 50%) were calculated according to a sigmoid dose–response curve using GraphPad Prism 5 software (GraphPad Software, Inc., La Jolla, CA, USA).

## Time-lapse imaging and analysis

A2780 and HCT116 cells (each, $5.0 \times 10^4$ per well) were cultured in the wells of 24-well culture plates and treated with T3 or DMSO. The treated cells were incubated for 3 h at 37°C under an atmosphere of 5% $CO_2$/95% air in a microscope stage-mounted humidified chamber (Incubator XL S1; Carl Zeiss AG, Oberkochen, Germany). Serial phase-contrast images were captured every 10 min using an Axiovert 200M microscope (Carl Zeiss AG) equipped with an EC Plan NeoFluar 20× lens (Carl Zeiss AG) and processed using AxioVision 4.5 software (Carl Zeiss AG), then compiled in JPEG format and exported to Adobe Photoshop for processing.

## Cell cycle analysis

A2780 and HCT116 cells (each $2.0 \times 10^5$ per well) were cultured in the wells of 6-well plates for specified periods of time, harvested, washed twice with ice-cold phosphate-buffered saline (PBS) and fixed in ice-cold 70% ethanol/30% distilled water (v/v), then washed twice in PBS and resuspended in Guava Cell Cycle reagent (Merck Millipore, Burlington, MA, USA) in accordance with the manufacturer's instructions. The DNA contents were determined using a FACScan™ automated flow cytometer with FlowJo software (BD Biosciences, Franklin Lakes, NJ, USA).

## Apoptosis analysis by staining with Annexin V and propidium iodide

A2780 and HCT116 cells (each $2.0 \times 10^5$ per well) were seeded into the wells of 6-well plates and incubated overnight with culture medium. The next day, diluted compounds were added to the culture media. At 24–48 h after incubation, the cells were harvested and stained with propidium iodide (PI) and FITC-conjugated Annexin V using the MEBCYTO® Apoptosis Kit (Medical & Biological Laboratories Co., Ltd., Nagoya, Japan) in accordance with the manufacturer's instructions. Annexin V binding and PI staining were analysed using the BD LSRFortessa™ cell analyser with FlowJo software (BD Biosciences).

## Caspase assay

A2780 ($8.0 \times 10^3$ per well) and HCT116 cells ($8.0 \times 10^3$ per well for 24-h treatment and $2.5 \times 10^3$ per well for 48-h treatment) were seeded into the wells 96-well plates and incubated

in culture medium. After overnight incubation, diluted compounds were added to the culture media. After 24–48 h of incubation, caspase-3/7 activity was measured using the Caspase-Glo 3/7 Assay System (Promega Corporation) in accordance with the manufacturer's instructions.

### Reverse transcriptase polymerase chain reaction (RT-PCR)

Total RNA was extracted using the RNeasy Plus Mini Kit (QIAGEN, Hilden, Germany) in accordance with the manufacturer's instructions. Reverse transcription was performed using the High-Capacity cDNA Reverse Transcription kit and AmpliTaq Gold 360 Master Mix (both were obtained from Thermo Fisher Scientific). The PCR conditions were as follows: *MCL1*: an initial denaturing step at 95˚C for 5 min, followed by 35 cycles at 94˚C for 30 s, 70˚C for 30 s and 72˚C for 45 s, and a final elongation step at 72˚C for 10 min; *BCL2L1*: 95˚C for 5 min, 35 cycles at 94˚C for 30 s, 67˚C for 30 s, 72˚C for 45 s and 72˚C for 10 min; and *ACTB*: 95˚C for 5 min, 35 cycles at 94˚C for 30 s, 60˚C for 30 s, 72˚C for 1 min and 72˚C for 10 min. The following primers were used for amplification (all 5' to 3'): MCL1-F: `CTCGGTAC CTTCGGGAGCAGGC`; MCL1-R: `CCAGCAGCACATTCCTGATGCC`; BCL2L1-F: `GAGGCAGGCG ACGAGTTTGAA`; BCL2L1-R: `TGGGAGGGTAGAGTGGATGGT`; cFLIP$_S$-F: `TAAGCTGTCTGTCG GGGACT`; cFLIP$_S$-R: `ATCAGGACAATGGGCATAGG`; cFLIP$_L$-F: `GGACCTTGTGGTTGAGTTGG`; cFLIP$_L$-R: `CATAGCCCAGGGAAGTGAAG`; ACTB-F: `CCAGCTCACCATGGATGATGATATCG`; and ACTB-R: `GGAGTTGAAGGTAGTTTCGTGGATGC`. The PCR products were separated by capillary electrophoresis and the signal intensity was measured using the LabChip GX Touch Nucleic Acid analyser (PerkinElmer, Inc.).

### Immunoblot analysis

Cells were harvested and lysed in Pierce IP Lysis Buffer (Thermo Fisher Scientific) supplemented with Halt Protease Inhibitor Cocktail (Thermo Fisher Scientific). After centrifugation of the lysates, the protein concentrations of the supernatants were determined using the Pierce BCA Protein Assay Kit (Thermo Fisher Scientific). The lysates were boiled for 5 min with Sample Buffer Solution with 3-mercapto-1, 2-propanediol (FUJIFILM Wako Pure Chemical Corporation, Tokyo, Japan). The samples were subjected to 5%–20% gradient sodium dodecyl sulphate-polyacrylamide gel electrophoresis. Once separated, the proteins were electrophoretically transferred to polyvinylidene fluoride polymer membranes, which were incubated with primary antibodies against Rictor (#2114), Raptor (#2280), Mcl-1 (#5453), Bcl-xL (#2762), Bcl-2 (#2870), XIAP (#2042), cFLIP (#56343), cIAP1 (#7065), cIAP2 (#3130), AKT (#9272) (Cell Signalling Technology, Danvers, MA, USA) and anti-vinculin (#CP74; Merck Millipore), and then incubated with species-specific HRP-conjugated secondary antibodies. SuperSignal West Femto Maximum Sensitivity Substrate (Thermo Fisher Scientific) or Amersham ECL Prime Western Blotting Detection Reagent (GE Healthcare, Chicago, IL, USA) was used for protein detection and images were captured digitally using the ImageQuant LAS4000 camera system (GE Healthcare).

### Statistical analysis

The number of biological replicates and statistical significance are noted in the figures, figure legends, and manuscript. Data are shown as mean ± standard deviation (SD). Statistical analysis and graph generation for cell growth assay and caspase assay were calculated using Graph-Pad Prism 5 software. Students's t test was used to determine the statistical significance for the caspase assay. Statistical significance was defined as $p < 0.05$.

## Results

### T3 induced apoptosis of A2780 cells and G2/M phase cell cycle arrest in HCT116 cells

Two representative human cancer cell lines (i.e., ovarian carcinoma A2780 and colorectal carcinoma HCT116 cells) were used to investigate the detailed mechanisms underlying the inhibition of cell growth and apoptosis induced by the splicing modulator T3 and its inactive analog T3-1. T3-1, with a fluoro group at 8-position, did not inhibit CLK1 or CLK2 (S1 Fig). The previous docking model of T3 [15], suggests that T3-1 might not be able to bind to the hinge region of CLK2. As shown in Fig 1A, T3 inhibited the growth of both A2780 and HCT116 cells ($GI_{50}$ = 345 and 122 nM, respectively), but T3-1 did not inhibit cell growth even at 10 μM (S1 Fig). Interestingly, T3 treatment induced apparently different morphological changes to A2780 and HCT116 cells (Fig 1B). A2780 cells began to shrink as early as 16–24 h of T3 treatment, whereas the cytoplasm content of HCT116 cells had increased for approximately 24 h. Cell cycle analysis revealed that T3 induced G2/M phase cell cycle arrest of HCT116 cells and apoptosis of A2780 cells after 16–24 h of treatment (Fig 1C). To examine the mechanism of apoptosis induction in more detail, Annexin V/PI and caspase-3/7 analyses were performed. The percentages of early apoptotic A2780 and HCT116 cells [Annexin V (+), PI (-)] after 24 h of treatment with 3 μM T3 were 29.1% and 3.2% (Fig 1D). After 48 h of treatment, 42.6% of the HCT116 cells were apoptotic. Next, caspase 3/7 activity was measured following T3 treatment. Caspase 3/7 activity was approximately five-fold greater in A2780 cells, as compared to HCT116 cells, which was consistent with the cell cycle and Annexin V data (Fig 1E). Additional treatment with the pan-caspase inhibitor Z-VAD-FMK (carbobenzoxy-valyl-alanyl-aspartyl-[O-methyl]- fluoromethylketone) suppressed up-regulation of caspase 3/7, suggesting that T3-induced apoptosis is caspase-dependent (Fig 1E). These results suggest that T3-induced apoptosis of A2780 and HCT116 cells was caspase-dependent and the mechanisms underlying cell growth inhibition and apoptosis induction differed between two types of cells.

### T3 repressed anti-apoptotic proteins IAP family proteins, cFLIP, and Mcl-1

Next, A2780 and HCT116 cells were used for examining whether T3 affected the expression of anti-apoptotic molecules. Bcl-2 was specifically expressed in A2780, whereas Bcl-xL and cIAP2 were specifically expressed in HCT116 (Fig 2). Mcl-1L and $cFLIP_L$ were highly expressed in A2780 compared with that in HCT116. T3 treatment for 16 h did not affect the protein levels of Bcl-2, Bcl-xL and AKT, but T3 decreased IAP family proteins (cIAP1, cIAP2 and XIAP) and Mcl-1L in a dose-dependent manner. Pro-apoptotic Mcl-1S was slightly expressed in A2780 and HCT116 cells. T3 modestly led to decrease of Mcl-1S, but the decreasing level was not remarkable compared to Mcl-1L. Mcl-1L was completely deleted after treatment with 3 μM T3 in HCT116 cells, whereas Mcl-1S was expressed in the same condition. In addition, T3 repressed two isoforms $cFLIP_L$ (55 kDa) and $cFLIP_S$ (25 kDa), whereas T3 induced a different size of $cFLIP_L$ and $cFLIP_S$ (Fig 2), suggesting that T3 altered the splicing of cFLIP. Taken together, these results indicated that T3 caused reduction of the levels of several anti-apoptotic proteins; however, no change was noted in Bcl-xL and Bcl-2 protein levels.

### T3 induced splicing alterations to the apoptosis-related gene MCL1 and CFLAR

Since each long and short isoform of Mcl-1 and Bcl-X have anti- apoptotic/pro-apoptotic function in cancer cells, A2780 and HCT116 cells were used to determine whether T3 altered splicing to the apoptosis-related *MCL1* and *BCL2L1* genes. As shown in Fig 3A, the pro-apoptotic

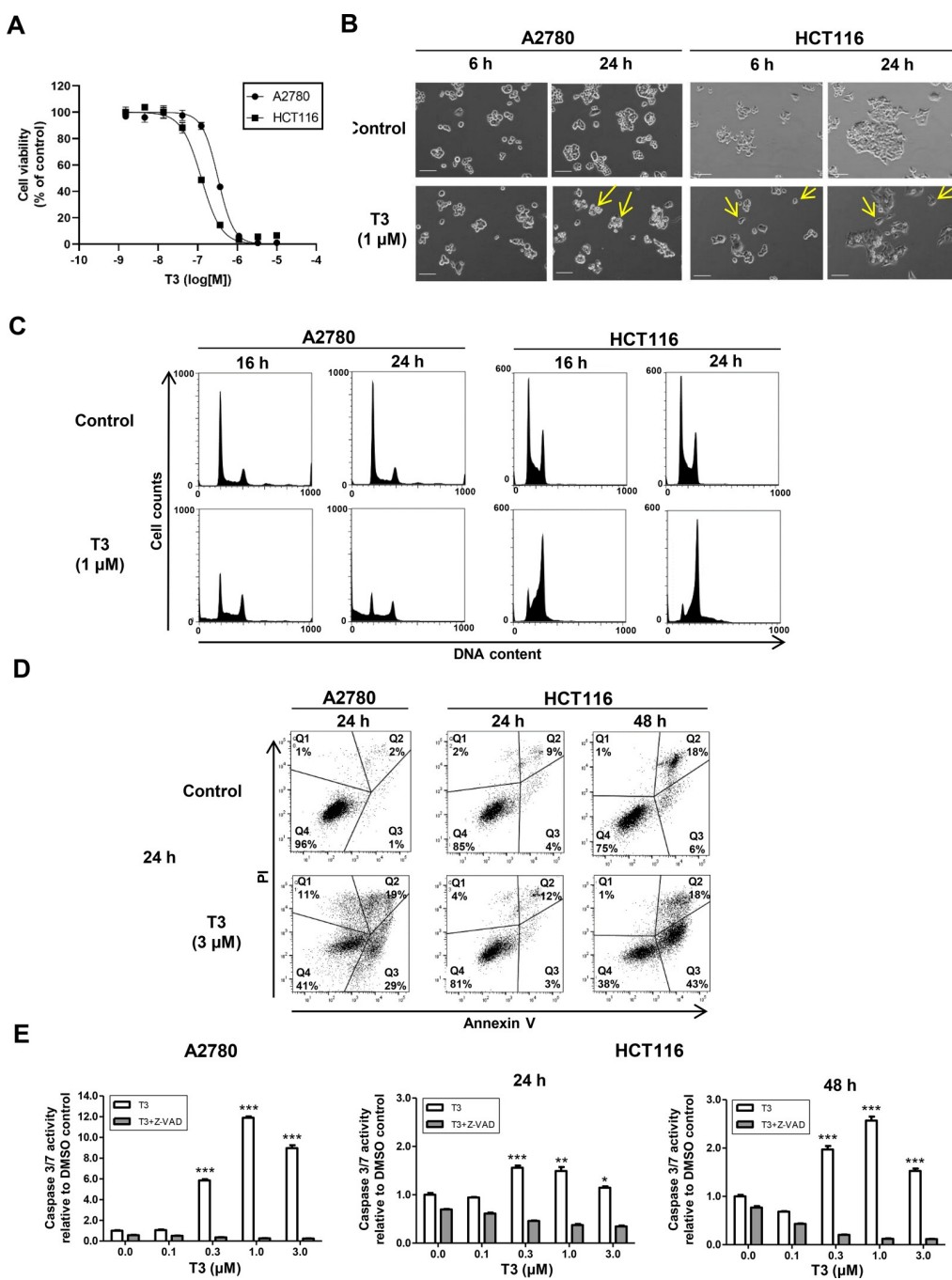

**Fig 1. T3 inhibits growth of HCT116 and A2780 cells via different mechanisms.** (A) Viabilities of A2780 and HCT116 cells normalized to the DMSO control after 72 h of T3 treatment. The x axis shows the compound concentration (logM). The y axis shows the percent inhibition of adenosine triphosphate contents, as compared to the DMSO control. Data are presented as the mean ± standard deviation (SD) of three independent experiments. (B) Time-lapse microscopic images of A2780 and HCT116 cells treated with DMSO or 1 μM T3 for 6 or 24 h. The yellow arrowheads show shrunken cells in A2780 and enlarged cytoplasm in HCT116 cells after treatment with T3 for 6 h or 24 h. Scale bar = 100 μm. (C) Flow cytometric analysis of the cell cycle distribution of HCT116 and A2780 cells treated with DMSO or 1 μM T3 for 16 and 24 h. Data are representative of two independent experiments. (D) Flow cytometric analysis of apoptosis of A2780 and HCT116 cells stained with Annexin V/PI. Cells were incubated with T3 for 24 h (A2780 and HCT116 cells) or 48 h (HCT116 cells). Apoptosis was analysed by Annexin V/PI staining and flow cytometry. The percentage of cells within each quadrant are indicated. Q1: Annexin V (-)/PI (+), Q2: Annexin V (+)/PI (+), Q3: Annexin V (+)/PI (-) and Q4: Annexin V (-)/PI (-). Data are representative of two independent experiments. (E) Caspase 3/7 activity was measured in the presence or absence of Z-VAD-FMK and is presented as the fold change, as

compared to the DMSO control samples. Data are presented at the mean ± SD of three independent experiments. Statistical analyses were performed using an unpaired Student's t-test (*P < 0.05; **P < 0.01; ***P < 0.001).

*MCL1S* isoform was dose-dependently induced by T3 treatment for 6 h and 16 h in both cell types, and the anti-apoptotic *MCL1L* isoform was downregulated by T3. The pro-apoptotic short isoform of *BCL2L1* (*BCL2L1S*) was induced by T3 at concentrations in the range of 0.1–1 µM for 16 h treatment in both cells; however, dose-dependent change was not observed. The "percent spliced in" value for the splicing of the *MCL1* and *BCl2L1* genes was calculated after T3 treatment. As shown in Fig 3B, T3 altered splicing of the anti-apoptotic isoform *MCL1L* to the pro-apoptotic isoform *MCL1S* at 6 and 16 h in a dose-dependent manner, but had no effect on splicing of *BCL2L1* in either cell type, regardless of the treatment duration. T3 and Z-VAD-FMK co-treatment induced the *MCL1S* isoform in both cell types, indicating that the

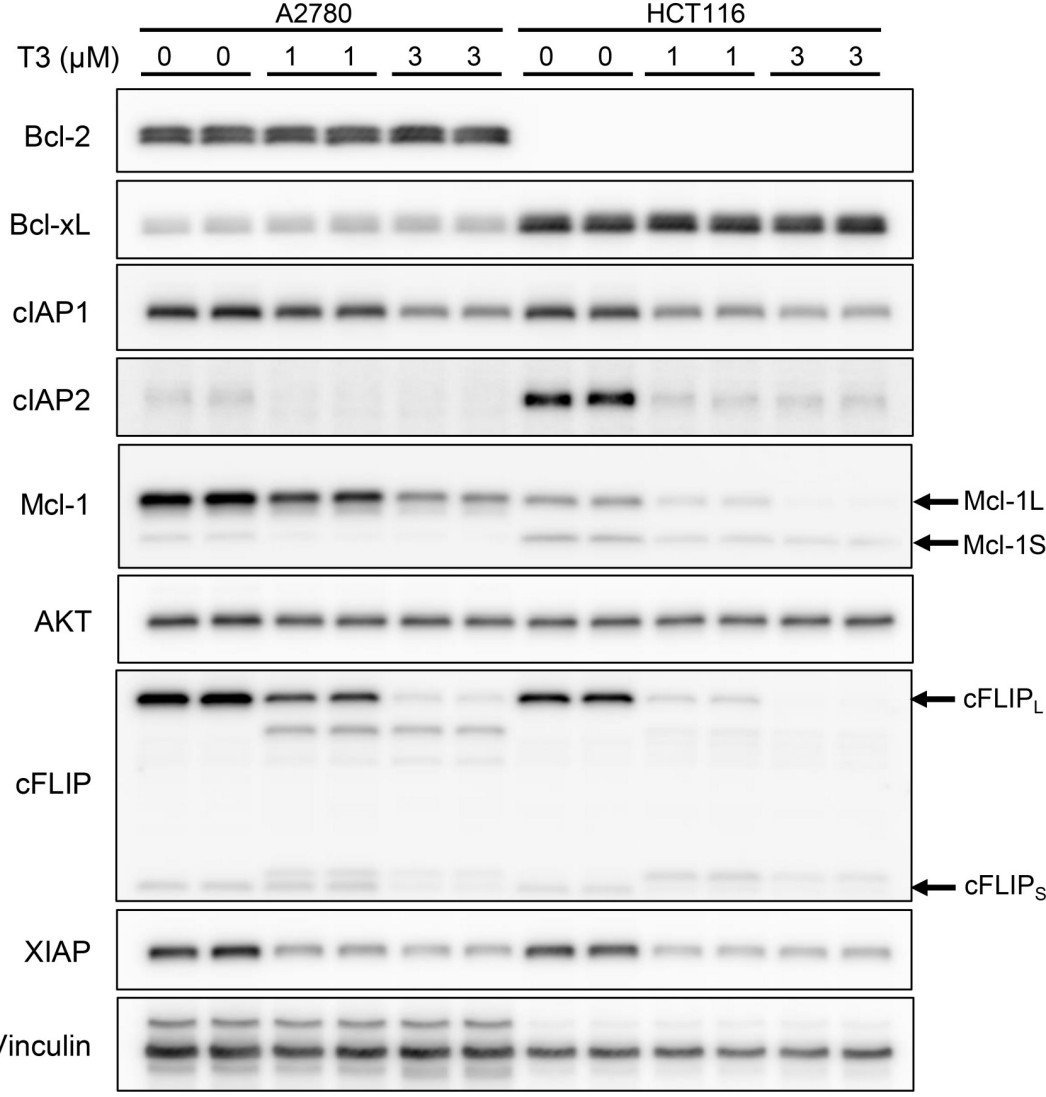

**Fig 2. T3-induced reduction of anti-apoptotic proteins expression in A2780 and HCT116 cells.** Immunoblot analyses were performed using protein lysates from A2780 and HCT116 cells treated with T3 for 16 h at the indicated concentrations. The positions of Mcl-1L, Mcl-1S, cFLIP$_L$, and cFLIP$_S$ are indicated on the right. Uncropped blot images are shown in S2 File.

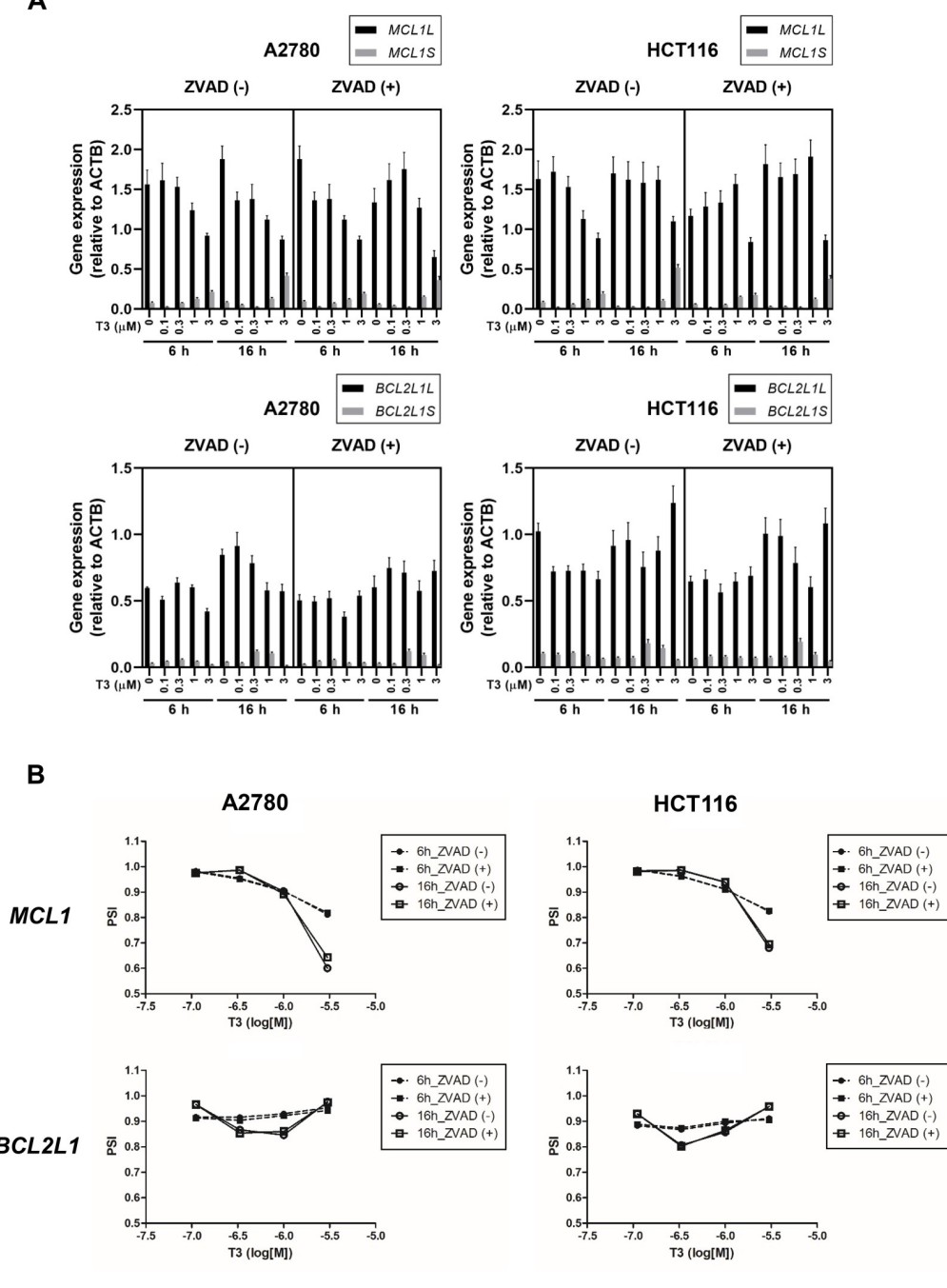

**Fig 3. Alteration of *MCL1* pre-mRNA splicing induced by T3 in A2780 and HCT116 cells.** (A) Cells were treated with T3 for 6 or 16 h at the indicated concentrations in the presence or absence of Z-VAD-FMK. Expression of *MCL1* and *BCL2L1* splice isoforms were analysed by RT-PCR, and PCR products were analysed by capillary electrophoresis. The quantitative expression for the long and short forms of *MCL1* and *BCL2L1*, respectively, are shown. *ACTB* mRNA expression was evaluated as an internal control. Data are representative of three independent experiments. (B) Signal intensity was quantified and the percent spliced in (PSI) values for *Mcl-1* and *BCL2L1* alternative exons after T3 treatment for 6 or 16 h in the presence or absence of Z-VAD-FMK are shown. Data are presented as the mean ± SD of three independent experiments.

up-regulation of *MCL1S* was not due to the indirect effect of caspase cleavage. In addition, we examined splicing of *CFLAR* long and short isoform (*CFLAR*-L and -S) after treatment with T3, since T3 induced a different size of cFLIP$_L$ and cFLIP$_S$ proteins as shown in Fig 2. RT-PCR analysis to detect *CFLAR*-L and *CFLAR*-S revealed that T3 decreased endogenously expressed *CFLAR*-L and *CFLAR*-S at 6 and 16 h in a dose-dependent manner, and T3 alternatively induced multiple smaller size of transcripts of *CFLAR*-L and *CFLAR*-S (Fig 4). These results suggest that T3 altered the splicing of *MCL1L* and *CFLAR* that has anti-apoptotic functions, but had no effect on splicing of the *BCL2L1* gene.

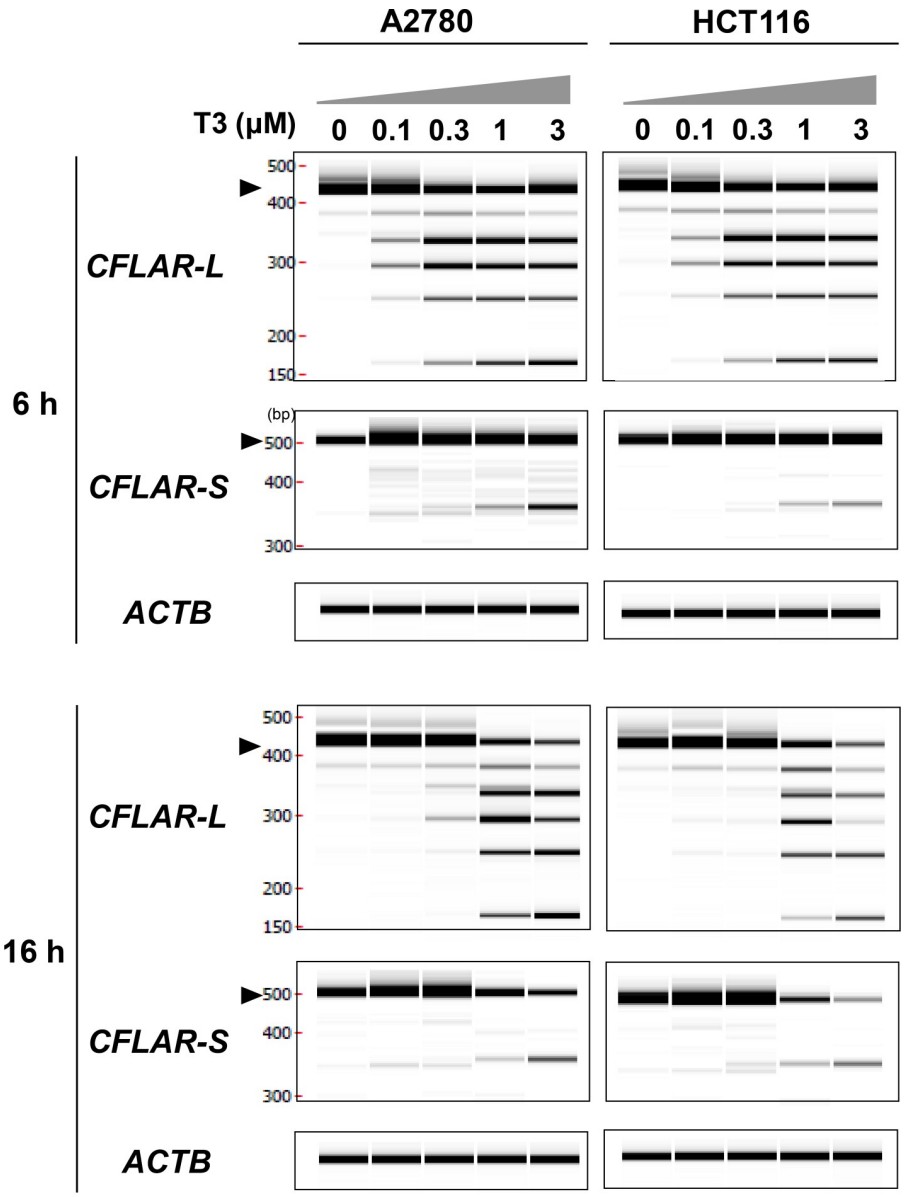

**Fig 4. Alteration of *CFLAR* pre-mRNA splicing induced by T3 in A2780 and HCT116 cells.** Cells were treated with T3 for 6 or 16 h at the indicated concentrations. Expression of *CFLAR* long and short splice isoforms (*CFLAR*-L and *CFLAR*-S) were analysed by RT-PCR, and PCR products were analysed by capillary electrophoresis. Closed triangles indicate the long and short forms of *CFLAR*-L, respectively. ACTB mRNA expression was evaluated as an internal control. Data are representative of three independent experiments. Raw electropherogram images are shown in the S3 File.

## T3 reduced the mammalian target of rapamycin (mTOR) component 2 (mTORC2) pathway

The Mcl-1 protein is stabilised by mTOR and rictor (mTORC2) via proteasomes and the ubiquitin pathway. Hence, inhibition of mTORC2 pathway leads to decreased Mcl-1 protein expression [21]. To examine whether the T3-induced reduction of Mcl-1 protein expression is regulated by splicing alterations or proteasome degradation, we examined mTORC2 pathway and proteasome regulation by treatment with T3. T3 treatment decreased Rictor expression more effectively than Raptor in A2780 and HCT116 cells (Fig 5A). Next, we investigated the possibility for the proteasome degradation with the use of the proteasome inhibitor MG-132. As shown in Fig 5B, MG-132 treatment without T3 induced Mcl-1L, not MCL1-S, protein production via blocking the proteasome pathway, while co-treatment with MG-132 and T3 decreased Mcl-1L and Mcl-1S protein levels in a T3-dose dependent manner. Additionally, MG-132 induced Mcl-1L and slightly induced Mcl-1S protein production even in the presence of 3 μM T3 (Fig 5B). These results imply that T3 degraded Mcl-1 through two independent pathways, splicing alterations of Mcl-1 and the mTORC2 proteasome pathway.

## T3 in combination with Bcl-2 inhibition synergistically induced apoptosis

Since alternative splicing and protein analysis revealed that T3-induced apoptosis was independent of Bcl-2 and Bcl-xL, the combined effect of T3 and the Bcl-xL/Bcl-2 inhibitor ABT-

**A**

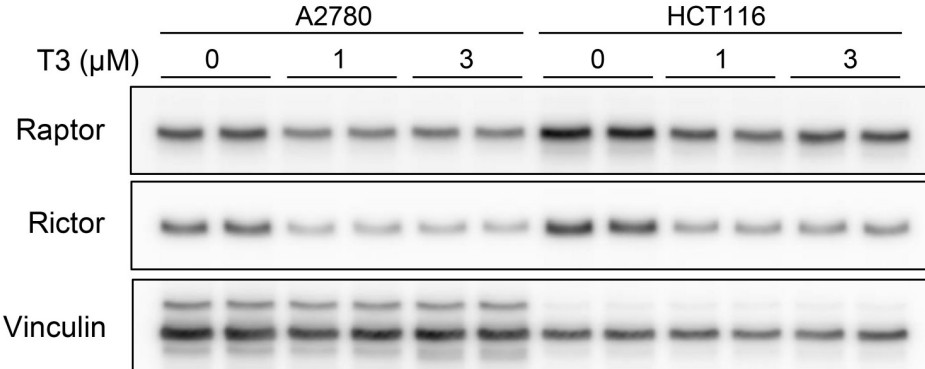

**B**

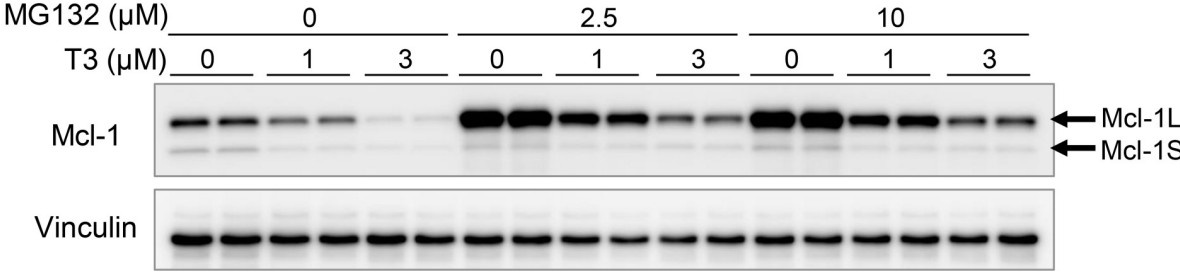

**Fig 5. T3-induced reduction of mTORC2 and involvement of proteasome pathway in A2780 and HCT116 cells.** (A) Immunoblot analyses were performed using protein lysates from A2780 and HCT116 cells treated with T3 for 16 h at the indicated concentrations. (B) HCT116 cells were pretreated with T3 for 16 h at the indicated concentrations and then co-treated with 2.5 or 10 μM of MG132 for 6 h. Data are representative of two independent experiments. (A) (B) Uncropped blot images are shown in S2 File.

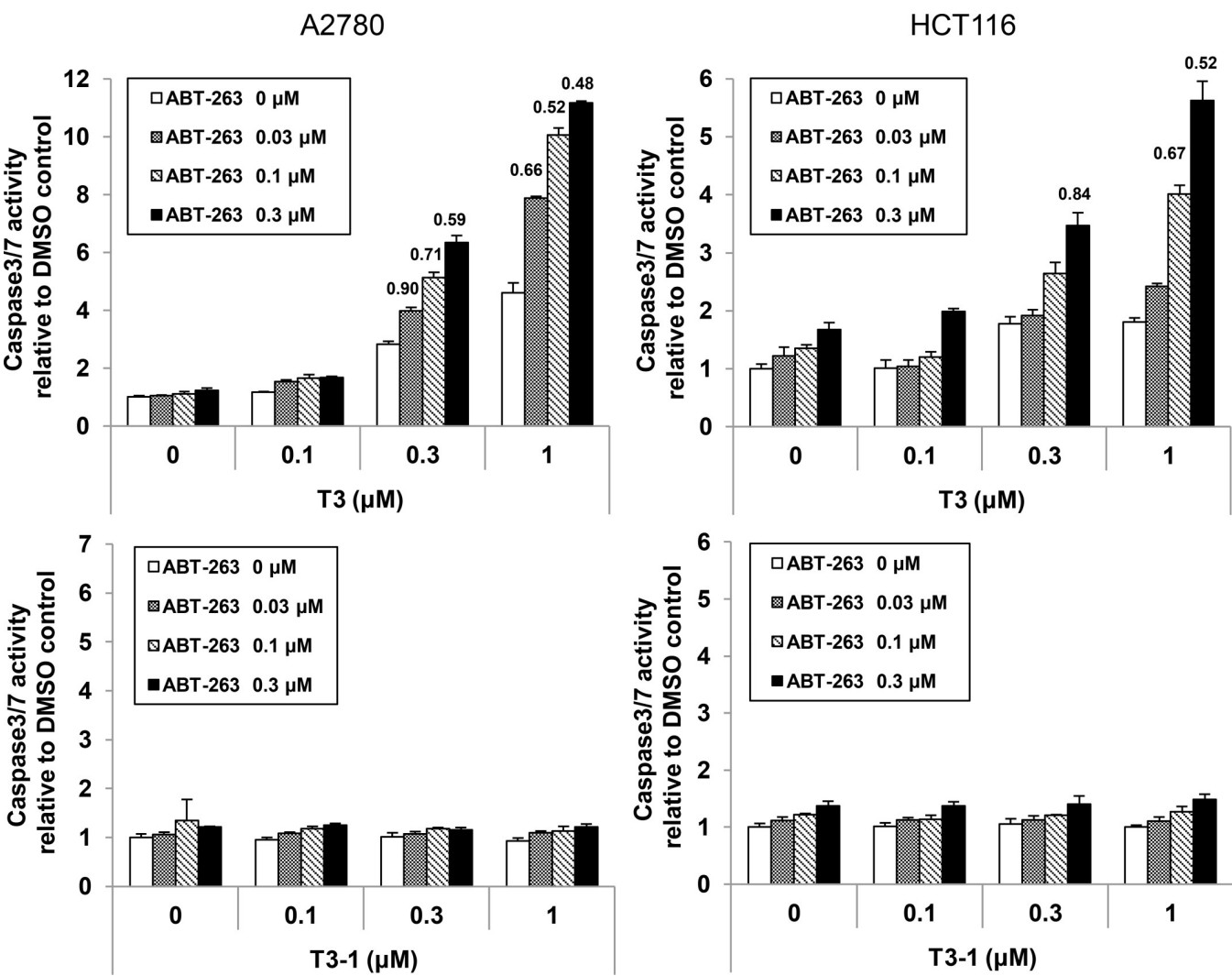

**Fig 6. Synergistic effects on apoptosis induction were observed by T3 in combination with Bcl-2 inhibitor in HCT116 and A2780 cells.** HCT116 and A2780 cells were treated with T3 or T3-1 and ABT-263 at the indicated concentrations for 16 h or 24 h, respectively. Caspase 3/7 activities were measured. Combination index (CI) values [24] are shown above the bar. A CI value of < 1 indicates synergism. Data are presented as the mean ± SD of two independent experiments.

263 was analysed [22,23]. The results showed that co-treatment with T3 and ABT-263 synergistically induced caspase 3/7 activation in A2780 and HCT116 cells (Fig 6). A synergistic effect was observed even with the combination of low-dose T3 and ABT-263 in A2780 cells, as compared to HCT116 cells. These results are consistent with our previous data from cell cycle and apoptosis induction analyses. In addition, caspase 3/7 activity was up-regulated by approximately six-fold after 16 h of co-treatment with 0.3 μM T3 and ABT-263 in A2780 cells (Fig 6). Single treatment with 0.3 μM T3 induced caspase 3/7 production by approximately three-fold at 16 h and six-fold at 24 h (Figs 1E and 6). Also, in HCT116 cells, caspase 3/7 was up-regulated to a similar level with T3 and ABT-263 co-treatment at 24 h and single T3 treatment at 48 h. T3-1, the inactive analog of T3, did not show synergistic effects with ABT-263 (Fig 6). These results suggest that T3 and ABT-263 co-treatment rapidly induced apoptosis, as compared to T3 treatment alone. Taken together, these results demonstrate that the combination of T3 and the Bcl-xL/Bcl-2 inhibitor promoted the death of cancer cells.

## Discussion

The use of splicing modulators targeting CLK or SF3b has recently gaining attention as a cancer treatment option since genetic mutations to spliceosome-related genes and splicing dysregulation have been found to inhibit cancer progression. It is of particular importance to clarify the molecular pathways involved in this drug combination strategy. The results of the present study showed that T3 led to decreased levels of several anti-apoptotic proteins for a short period of exposure and altered splicing of Mcl-1 and cFLIP, but not Bcl-xL. Consistent results were obtained from the co-treatment analysis, indicating that the combination of T3 with the Bcl-xL/Bcl-2 inhibitor synergistically induced apoptosis. These data provide convincing evidence that the combination of T3 and the Bcl-xL inhibitor is an appropriate molecular-based combination for cancer treatment. The SF3b inhibitor in combination with Bcl-xL inhibitors also has a synergistic effect [15,25]. The molecular mechanisms underlying splicing alterations presumably differ, although the use of CLK and SF3b inhibitors induced splicing alterations to multiple genes. CLKs recognise exons via phosphorylation of serine/arginine-rich proteins and mainly induces exon skipping [14,15]. On the contrary, SF3b complex inhibitors preferentially induced splicing within the introns [26,27]. It is conceivable that crosstalk between CLK and the SF3b pathway occur in alternative splicing of several genes, including *MCL1*.

T3 switched from the splicing of *MCL1L* isoform to *MCL1S* in mRNA levels. However, T3 reduced Mcl-1S as well as Mcl-1L in protein levels, though decreasing level of Mcl-1S was not significant compared to Mcl-1L. It is conceivable that T3 degrades Mcl-1 proteins including long and short form via mTORC2-proteosome pathway.

Mcl-1 up-regulation is a hallmark of cancers and causes drug resistance to some cancer therapies [28,29]. For instance, treatment of tumours with the multi-tyrosine kinase inhibitor sunitinib enhances the stability of Mcl-1 and induces activation of the mTORC pathway, which leads to sunitinib resistance [28]. Since CLK inhibitor decreases the expression levels of Mcl-1 and S6K [14], CLK inhibitors are viable therapeutic options for the treatment of sunitinib-resistant cancers. The study results also showed that T3 induced apoptosis of A2780 cells and cell cycle arrest at the G2/M phase in HCT116 cells. Mcl-1 is highly expressed in A2780 cells, as compared to that in HCT116 cells. Additionally, the depletion of Mcl-1 induced apoptosis of A2780 cells, whereas apoptosis of HCT116 cells required the depletion of all Bcl-2 family members (i.e., Mcl-1, BCL2 and Bcl-xL) [30,31]. These findings suggest that higher Mcl-1 expression can be induced by the combination treatment with CLK and Bcl-xL/Bcl-2 inhibitors, although further studies with additional cell types are needed to confirm these results.

In addition to Mcl-1, the levels of other members of anti-apoptotic proteins, IAP family proteins (cIAP1, cIAP2 XIAP) and cFLIP were decreased by T3 treatment in both A2780 and HCT116 cells. Gene ontology analysis of T3-responsive comprehensive alternative splicing events revealed that the apoptotic pathways were enriched by 0.05–10 μM of T3 exposure in HCT116 cells [15]. However, the splicing alterations of IAP family were not observed in this RNA-seq data set. It is conceivable that the T3-responsive reductions of IAP family proteins were induced by unknown RNA processing machinery, such as transcriptional control and nuclear-cytoplasmic mRNA transport, and not splicing modification. As shown in Figs 2 and 4, T3 resulted in splicing alterations of *CFLAR*-L and *CFLAR*-S, and subsequent reduction of canonical cFLIP-L and cFLIP-S proteins. Spliced isoforms with premature termination codons induced by CLK inhibitor are degraded by stimulation of nonsense-mediated mRNA decay [14,15]. Presumably, T3-induced novel isoform proteins were detected using anti-cFLIP antibody in this study, although further sequence-based analysis is required.

MYC amplification and high CLK2 expression in cancer cells are considered predictive of the response to a CLK inhibitor [20]. Drug combination strategies with CLK and Bcl-xL

inhibitors might be effective in combination with these predictive biomarkers. Taken together, the results of this study showed that the reductions of anti-apoptotic proteins and alterations to *MCL1* splicing is an important mechanism of CLK-induced cell death and the use of CLK inhibitors in combination with Bcl-xL/Bcl-2 inhibitors presents an option for cancer therapy.

## Supporting information

**S1 Fig. Structures of T3 and T3-1, and cell growth inhibitory activity of T3-1.** (A) Chemical structures of T3 and T3-1. (B) Viabilities of A2780 and HCT116 cells normalized to the DMSO control after 72 h of T3 treatment. The x-axis shows compound concentration (logM). The y-axis shows percent inhibition of adenosine triphosphate content, compared to the DMSO control. Data are presented as mean ± standard deviation (SD) of three independent experiments.
(PDF)

**S1 File. Synthesis of T3-1.**
(PDF)

**S2 File. Uncropped blot images for Figs 2, 5A and 5B.**
(PDF)

**S3 File. Raw electropherogram images using the capillary electrophoresis LabChip for Fig 4.**
(PDF)

## Acknowledgments

We thank H. Miyake, Y. Ebisuno, S. Kondo, Y. Kikukawa and T. Hoshino for their support of this research.

## Author Contributions

**Conceptualization:** Aiko Murai, Shinsuke Araki.

**Data curation:** Aiko Murai, Shunsuke Ebara, Shinsuke Araki.

**Formal analysis:** Aiko Murai, Shunsuke Ebara.

**Investigation:** Aiko Murai, Toshiyuki Nomura.

**Methodology:** Aiko Murai, Shunsuke Ebara, Satoshi Sasaki, Tomohiro Ohashi, Tohru Miyazaki.

**Resources:** Satoshi Sasaki, Tomohiro Ohashi, Tohru Miyazaki.

**Supervision:** Toshiyuki Nomura, Shinsuke Araki.

**Validation:** Aiko Murai.

**Writing – original draft:** Aiko Murai, Shinsuke Araki.

**Writing – review & editing:** Shinsuke Araki.

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
