## [Decision Letter · Decision Letter 0]

4 Mar 2020

PONE-D-20-00375

Synergistic apoptotic effects in cancer cells by the combination of CLK and Bcl-2 family inhibitors

PLOS ONE

Dear Dr. Araki,

Thank you for submitting your manuscript to PLOS ONE. After careful consideration, we feel that it has merit but does not fully meet PLOS ONE’s publication criteria as it currently stands. Therefore, we invite you to submit a revised version of the manuscript that addresses the points raised during the review process.

Additional experimental data are required to confirm the hypothesis. The specificity of T3 should be confirmed. More data are required to prove that the phenotypes observed upon treatment are CLK- or mRNA splicing-dependent. Non-cancerous cells should be tested.

We would appreciate receiving your revised manuscript by Apr 18 2020 11:59PM. To enhance the reproducibility of your results, we recommend that if applicable you deposit your laboratory protocols in protocols.io, where a protocol can be assigned its own identifier (DOI) such that it can be cited independently in the future. For instructions see: http://journals.plos.org/plosone/s/submission-guidelines#loc-laboratory-protocols

We look forward to receiving your revised manuscript.

Kind regards,

Irina V. Lebedeva, Ph.D.

Academic Editor

PLOS ONE

Journal Requirements:

2. Please provide additional information about each of the cell lines used in this work, including any quality control testing procedures (authentication, characterisation, and mycoplasma testing). For more information, please see http://journals.plos.org/plosone/s/submission-guidelines#loc-cell-lines.

3. Thank you for providing the following Funding Statement: 

"Takeda Pharmaceutical Company Limited provided support in the form of salaries for authors but did not have any additional role in the study design, data collection and analysis, decision to publish, or preparation of the manuscript.".

We note that one or more of the authors is affiliated with the funding organization, indicating the funder may have had some role in the design, data collection, analysis or preparation of your manuscript for publication; in other words, the funder played an indirect role through the participation of the co-authors. 

* If the funding organization did not play a role in the study design, data collection and analysis, decision to publish, or preparation of the manuscript and only provided financial support in the form of authors' salaries and/or research materials, please review your statements relating to the author contributions, and ensure you have specifically and accurately indicated the role(s) that these authors had in your study in the Author Contributions section of the online submission form. Please make any necessary amendments directly within this section of the online submission form.  Please also update your Funding Statement to include the following statement: “The funder provided support in the form of salaries for authors [insert relevant initials], but did not have any additional role in the study design, data collection and analysis, decision to publish, or preparation of the manuscript. The specific roles of these authors are articulated in the ‘author contributions’ section.”

If the funding organization did have an additional role, please state and explain that role within your Funding Statement. 

* Please also provide an updated Competing Interests Statement declaring this commercial affiliation along with any other relevant declarations relating to employment, consultancy, patents, products in development, or marketed products, etc.  

Reviewers' comments:

Reviewer's Responses to Questions

**Comments to the Author**

1. Is the manuscript technically sound, and do the data support the conclusions?

Reviewer #1: Yes

Reviewer #2: Partly

2. Has the statistical analysis been performed appropriately and rigorously? 

Reviewer #1: Yes

Reviewer #2: N/A

3. Have the authors made all data underlying the findings in their manuscript fully available?

Reviewer #1: Yes

Reviewer #2: No

4. Is the manuscript presented in an intelligible fashion and written in standard English?

Reviewer #1: Yes

Reviewer #2: Yes

5. Review Comments to the Author

Reviewer #1: The manuscript of Murai and co-authors examined the combined effects of a splicing inhibitor with an inhibitor of apoptosis for synergy in cell models of ovarian (A2780) and colorectal cancer (HCT116). In this study they present evidence that the putative CLK3 inhibitor T3 results in inhibition of splicing of anti-apoptotic proteins. Combined with an BCL-2 inhibitor (ABT-263), the authors provide evidence for synergistic increases in cell death as determined by caspase 3 assays. The authors provide further evidence that, in part, these effects are mediated through reduction of mTor signaling.

Critique

Overall these are carefully performed studies that provide initial evidence for a combinatorial approach to improve induction of cell death in cancer cells. However, there are a number of issues that need to be addressed before the manuscript is acceptable for publication.

1. Only one CLK inhibitor was examined and the specificity of T3 was not discussed. The authors need to add additional experiments with another CLK inhibitor to determine if this is a drug, or target specific response. Evidence of the selectivity of T3, with respect to other kinases, needs to be provided. These studies would be more convincing if synergy was observed with other CLK inhibitors as well.

2. The inhibition of mTOR signaling is suggested as a key part of the response although this is not well validated. Additional data demonstrating this (i.e. inhibition of S6 phos, EIF4EBP phos, or other) should be provided. At present the authors discuss the loss of S6- but no data is actually shown.

3. The explanation for the loss of MCL1 is not clear (Fig. 3C). In particular, the effects of MG132 are quite minor. Based on the PCR data shown, its possible that T3 is blocking total MCL1 expression. While the one splice (splice in) is shown, the total MCL1 message should also be presented. The potential effects of T3 on pTEF/transcriptional regulation of MCL1 have not been ruled out.

Reviewer #2: Authors showed that the small molecule splicing modulator T3, a CDC-like kinase (CLK) inhibitor, induced cell cycle changes and apoptosis in tumor cell lines A2780 and HCT116. T3 also showed synergistic pro-apoptotic effect with ABT-263. Authors also observed changes at protein levels for several apoptotic proteins following T3 treatment. This is an interesting study and has a potential in advancing cancer therapy. However, in the current version of this manuscript, there is a lack of direct evidence showing that the phenotypes are indeed CLK- or mRNA splicing-dependent. Limited studies on the changes in protein levels and the gene expression of only selected isoforms are not sufficient to reach the conclusion that the alternative splicing plays a critical role. Therefore, further studies are warranted before publication.

1. Authors need to directly show that the phenotypes induced by T3 are CLK- or mRNA splicing-dependent. For example, cells with mutant CLK (unable to bind to T3, or splicing function deficient), or CLK knockout, or through any other means to show that T3-induced cell cycle changes, apoptosis, and synergy with ABT-263 are CLK- or mRNA splicing-dependent. Comparing inactive T3 analog-induced vs T3–induced phenotypic changes would further help understanding the underlying mechanism.

2. Direct evidence at mRNA levels by such as RT-PCR showing alteration of splicing events after T3 treatment should be provided for each of the splicing variants, including cFLIP-L, cFLIP-S, Mcl-1L, and Mcl-1S. Whether the mRNA splicing of those apoptotic proteins is CLK dependent also needs to be investigated.

3. Authors claimed that some of the Bcl-2 family members such as Bcl-XL and Bcl-2 were resistant to T3-induced splicing modulation. Have authors investigated other isoforms of those proteins such as Bcl-xS, Bcl2L2, etc, to support the claim?

4. In Figure 3A, authors claimed “T3 altered splicing of the anti-apoptotic isoform MCL1L to the pro-apoptotic isoform MCL1S at 6 and 16 h in a dose-dependent manner”, however, no data for MCL-1S has been shown. PSI for both MCL-1L and MCL-1S need to be showed.

5. In Figure 3C, both western blot and RT-PCR for total MCl-1 and its isoforms Mcl-1L and Mcl-1S should be shown, in order to enable the comparison and delineate the effect of T3 on splicing alteration.

6. A section of statistical analysis should be included.

6. PLOS authors have the option to publish the peer review history of their article (what does this mean?). If published, this will include your full peer review and any attached files.

Reviewer #1: No

Reviewer #2: No

---

## [Author Response · Author response to Decision Letter 0]

22 Apr 2020

We thank the academic editor and the reviewers for their constructive comments and suggestions. Following the requests from the editor and reviewers, we have performed additional experiments and included descriptions of these experiments in the Introduction, Results, and Discussion sections. We believe that the manuscript has been strengthened through this process. We have responded to each of the reviewers’ comments on the 'Response to reviewer' file.

We hope that the attached responses and the changes we have made are sufficient to make our manuscript suitable for publication in PLOS ONE. Please do not hesitate to contact me if you have any further queries, or to let me know if I can be of further assistance.

---

## [Decision Letter · Decision Letter 1]

24 Jun 2020

PONE-D-20-00375R1

Synergistic apoptotic effects in cancer cells by the combination of CLK and Bcl-2 family inhibitors

PLOS ONE

Dear Dr. Araki,

Thank you for submitting your manuscript to PLOS ONE. After careful consideration, we feel that it has merit but does not fully meet PLOS ONE’s publication criteria as it currently stands. Therefore, we invite you to submit a revised version of the manuscript that addresses the points raised during the review process.

Please address the Reviewer 2 concerns

We look forward to receiving your revised manuscript.

Kind regards,

Irina V. Lebedeva, Ph.D.

Academic Editor

PLOS ONE

Reviewers' comments:

Reviewer's Responses to Questions

**Comments to the Author**

1. If the authors have adequately addressed your comments raised in a previous round of review and you feel that this manuscript is now acceptable for publication, you may indicate that here to bypass the “Comments to the Author” section, enter your conflict of interest statement in the “Confidential to Editor” section, and submit your "Accept" recommendation.

Reviewer #1: (No Response)

Reviewer #2: (No Response)

2. Is the manuscript technically sound, and do the data support the conclusions?

Reviewer #1: Partly

Reviewer #2: Partly

3. Has the statistical analysis been performed appropriately and rigorously? 

Reviewer #1: Yes

Reviewer #2: Yes

4. Have the authors made all data underlying the findings in their manuscript fully available?

Reviewer #1: Yes

Reviewer #2: No

5. Is the manuscript presented in an intelligible fashion and written in standard English?

Reviewer #1: No

Reviewer #2: Yes

6. Review Comments to the Author

Reviewer #1: Please revise sentence on p.4 line 1-does not currently make sense.

The digitized blots in Figs 3 and 4 are unacceptable.

Reviewer #2: Authors did not directly and experimentally address many major concerns the reviewer has raised. One major point is the specificity aspect. Even though T3 has been reported to be potent, the off-target binding is still a possibility unless proven otherwise. Both the reviewers have specifically indicated that additional experiments with pharmacological and/or genetic approaches should be performed in order to show that the effect is CLK- or splicing-dependent. Unfortunately, authors did not attempt to address this point experimentally.

Figure 3A showed that the lower molecular weight BCL2L1 isoform Bcl-xS increased dose-dependently upon T3 treatment up to 1 uM, which is especially apparent after 16h treatment in both cell lines, and the amount of change is similar to MCL1 but with a different kinetics. However, authors described T3 “had no effect on splicing of the BCL2L1 gene”. This causes confusion.

There is also a considerable confusion regarding cFLIP. Authors stated that “T3 induced smaller size of cFLIPL and larger size of cFLIPL”, and then “T3 induced the smaller sized cFLIPL and cFLIPS”, and also stated “T3 alternatively induced multiple smaller sized transcripts of CFLAR-L and CFLAR-S”. In fact, the Fig 2 showed that T3 induced LARGER size of cFLIPs protein, but in Fig 4 there is NO LARGER size of cFLAR-S RNA but only SMALLER size cFLAR-S RNA. Would authors be able to consolidate?

Upon T3 treatment, Mcl-1S increased at RNA level (Fig 3A) but decreased at protein level (Fig 2). Reviewer’s Point 5 suggested to study the MCl-1 isoforms in Fig 3C (the original version), with using MG132 to investigate if the discrepancy in RNA and protein levels is at least due to proteasome degradation. Would the levels of Mcl-1S RNA become consistent with its protein levels if MG132 is present? Unfortunately, authors did not address this point experimentally.

7. PLOS authors have the option to publish the peer review history of their article (what does this mean?). If published, this will include your full peer review and any attached files.

Reviewer #1: No

Reviewer #2: No

---

## [Author Response · Author response to Decision Letter 1]

14 Sep 2020

We thank the academic editor and reviewers for their constructive comments and suggestions. According to the suggestions provided by the editor and reviewers, we performed additional experiments and included descriptions of these experiments in the Introduction, Results, and Discussion sections. The reviewers’ comments and our responses are listed in the attached file"Responses to reviewers".

---

## [Decision Letter · Decision Letter 2]

2 Oct 2020

Synergistic apoptotic effects in cancer cells by the combination of CLK and Bcl-2 family inhibitors

PONE-D-20-00375R2

Dear Dr. Araki,

We’re pleased to inform you that your manuscript has been judged scientifically suitable for publication and will be formally accepted for publication once it meets all outstanding technical requirements.

Kind regards,

Irina V. Lebedeva, Ph.D.

Academic Editor

PLOS ONE

Additional Editor Comments (optional):

Reviewers' comments:

Reviewer's Responses to Questions

**Comments to the Author**

1. If the authors have adequately addressed your comments raised in a previous round of review and you feel that this manuscript is now acceptable for publication, you may indicate that here to bypass the “Comments to the Author” section, enter your conflict of interest statement in the “Confidential to Editor” section, and submit your "Accept" recommendation.

Reviewer #1: All comments have been addressed

Reviewer #2: All comments have been addressed

2. Is the manuscript technically sound, and do the data support the conclusions?

Reviewer #1: Yes

Reviewer #2: Yes

3. Has the statistical analysis been performed appropriately and rigorously? 

Reviewer #1: Yes

Reviewer #2: Yes

4. Have the authors made all data underlying the findings in their manuscript fully available?

Reviewer #1: Yes

Reviewer #2: Yes

5. Is the manuscript presented in an intelligible fashion and written in standard English?

Reviewer #1: Yes

Reviewer #2: Yes

6. Review Comments to the Author

Reviewer #1: No further changes necessary- although I am still not pleased with Figure 4. Otherwise the authors have done a good job responding to my critiques.

Reviewer #2: (No Response)

7. PLOS authors have the option to publish the peer review history of their article (what does this mean?). If published, this will include your full peer review and any attached files.

Reviewer #1: No

Reviewer #2: No

---

## [Editor Report · Acceptance letter]

6 Oct 2020

PONE-D-20-00375R2 

Synergistic apoptotic effects in cancer cells by the combination of CLK and Bcl-2 family inhibitors 

Dear Dr. Araki:

I'm pleased to inform you that your manuscript has been deemed suitable for publication in PLOS ONE. Congratulations! Your manuscript is now with our production department. 

Kind regards, 

on behalf of

Dr. Irina V. Lebedeva 

Academic Editor

PLOS ONE